# Fields Touched by Digitalization: Analysis of Scientific Activity in Scopus

**Lorena Espina-Romero** [1,*] and **Jesús Guerrero-Alcedo** [2]

1   Escuela de Postgrado, Universidad San Ignacio de Loyola, La Molina, Lima 15024, Peru
2   Carrera de Psicología, Universidad Científica del Sur, Lima 15067, Peru
*   Correspondence: lespina@usil.edu.pe

**Abstract:** This study aims to analyze the publications in Scopus around digitalization in the space of time between 2018 and 2022. A bibliometric review is carried out with a bibliographic approach for 658 documents, which were processed by RStudio and VOSviewer software. The findings show the ten fields where digitization is most applied: "Archives, Corruption and Economy", "Industry 4.0, Internet of Things, Sustainability and Big Data", "Cultural Heritage, Deep Learning, Preservation and BIM", "Photogrammetry and 3D Digitalization", "Artificial Intelligence (AI) and Supply Chain Management", "Augmented Reality, Machine Learning and Virtual Reality", "Innovation, Business Model and Publishing Industry", "Algorithms, E-government and Biometrics", "Digital Collections" and "Healthcare". It should be noted that this document is based on 88.14% original studies, validating the results obtained, and it is also one of the most updated studies.

**Keywords:** sustainability; augmented reality; business model; companies; economy; machine learning; virtual reality





## 1. Introduction

Digitization can be defined as the action of moving physical elements and analogous methods to the digital plane, which involves the consideration of the use of data warehouses or the scanning of paper files to record all relevant documents, discarding outdated filing cabinets. It may be implemented as a system at the service of the informative, investigative and managerial legacy of any organization since it facilitates the supervision of a volume of information, as well as its editing and management [1–3].

Digitalization has spread across several fields transversally regardless of their nature. Its development advances according to the movement around sustainability, bringing as a benefit the improvement of a field's environmental impact [4–6]. Thanks to digitalization, it has been possible to reduce the use of paper and transport; additionally, sustainable materials and renewable energies are within reach [7–9].

It should be noted that digitalization helps to meet the Sustainable Development Goals (SDGs) since it can be useful for reducing poverty and hunger, promoting health, the creation of new jobs, mitigating climate change, increasing energy efficiency and making cities and communities more sustainable [10].

Digitalization is applied in numerous areas, one of these being human resources, which leads to the transformation of the operational management of a department for its automation and storage [11]. It is also applied in medicine for health care, giving patients the ability to communicate with their doctor via teleconference; that is, they are given full access to medical care without affecting their work schedule or leaving the tranquility of their home [12].

Additionally, digitalization is applied in public administration [13], where it is made up of a package of activities focused on the development and updating of processes, integrated by the reengineering of procedures and policies and their automation [14]. At the same

time, it is applied in companies, and in this area refers to the process by which a company uses materials, techniques and digital environments in order to provide significant benefits to users, solutions, new experiences and business models [15].

The Scopus database represents the first research conducted on digitization developed by [16], which was initially oriented to the medical area and aimed to provide a procedure with which to achieve the rehabilitation of hands after the amputation of five fingers. There are numerous publications on digitalization, many of them closely related to the variables "analog conversion" [17], "image processing" [18], "digital libraries" [19], "algorithms" [20], "automation" [21], "digital storage" [22], "information management" [23], "cultural heritage" [24], "digital transformation" [25], "artificial intelligence" [26] and "digital technologies" [27], among others.

After a review of the literature, it is possible to find several recent "review"-type studies linked to digitalization [28]. Among them is one prepared by [29], which is oriented towards the sustainable operational effectiveness of maritime transport. It is followed by the document developed by [30], which is focused on the digitalization and automation of agriculture taking advantage of the benefits of the joint work between the Internet of Things and Artificial Intelligence. There is also the manuscript prepared by [31], which tries to extract the points of view, intellectual framework and theoretical evolution of sustainability at the community level. Finally, the document compiled by [32] is presented, where they show how Calpurnio and El Roto (two Spanish authors) encourage critical reflection on the digital collective with graphic satire.

As shown in the previous paragraph, each of the cited documents focuses on a specific field within the study of digitization, but none make a general mapping of the fields it touched. For these reasons, this study is important because it addresses these fields in a general way through an evaluation of the scientific activity that is carried out in this area, and thus is able to present the scope of digitization. In this sense, this study is formulated with the following research question:

RQ1. What are the fields of study where digitalization is most applied?

To answer this question, the central objective of this bibliometric review is to analyze the publications around digitization in the Scopus database, using 2018–2022 as its space of time. Therefore, the motivation of this study is to identify the fields touched by digitization, as well as to detail lines of research rarely explored. It should be noted that this bibliometric study is accompanied by a bibliographic approach to acknowledge the opinions of the authors involved in this study. Before responding to RQ1, some key information about the publications selected for this study is briefly given, and, after RQ1, a concise agenda for future research is concluded.

## 2. Methodology

This study is based on a bibliometric review, which is applied in various scientific fields acquiring validity among academics [33]. In this bibliometric study, the five steps proposed by Zupic and Carter [34] were applied, including: (1) study design, (2) bibliometric data collection, (3) analysis, (4) visualization and (5) interpretation (Figure 1).

**5 steps of Zupic and Carter (2015)**

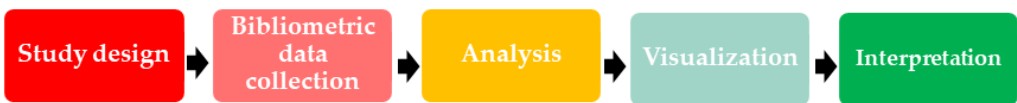

**Figure 1.** Methodological design.

### 2.1. Study Design

After the literature review, the following research question was asked: What are the fields of study where digitization is most applied? To answer the question, the bibliometric

method of conceptual structure called Thematic Map was chosen, which indicates the directions of the questioned fields. To extract the fields of study rarely explored (future research agenda), the bibliometric method called the "analysis of co-occurrence of keywords" was chosen, which indicates the keywords of lower co-occurrence, allowing us to infer the fields less touched by digitization.

### 2.2. Collection of Bibliometric Data

Publications whose titles contained the keyword "digitization" were selected in Scopus, obtaining 4175 documents. Open access documents were selected, obtaining 1069. Subsequently, the documents of the period 2018–2022 were selected, obtaining 658. This period was selected because the purpose of this study is to analyze current studies and not past literature. Consequently, the following search string was generated: "TITLE (digitization) AND (LIM-IT-TO (OA, "all")) AND (LIMIT-TO (PUBYEAR, 2022) OR LIMIT-TO (PUBYEAR, 2021) OR LIMIT-IT- A (PUBYEAR, 2020) OR LIMIT-TO (PUBYEAR, 2019) OR LIMIT-TO (PUBYEAR, 2018))". All types of documents, all countries and all areas of knowledge were included.

### 2.3. Analysis

In this step, the data were loaded and their conversion was achieved to maintain their viability and quality. The data were obtained from Scopus in RIS, BibTex and CSV format for later loading into VOSviewer software version 1.6.18 (Leiden, The Netherlands), RStudio version R 4.1.1 (Vienna, Austria) and the Microsoft Excel 365 web application (Redmond, WA, USA).

### 2.4. Visualization

This study was based on the Thematic Map generated by RStudio software to achieve the visualization of the fields of study most affected by digitalization. Additionally, it relied on the "analysis of co-occurrence of keywords" generated by VOSviewer software in order to visualize the possible fields little explored (future research agenda). Finally, it relied on graphs generated by the Microsoft Excel 365 web application to display the main information (documents by year, countries and thematic area).

### 2.5. Interpretation

This step describes the findings and gives an appropriate interpretation. In line with the research question of this study, the results shown in the main information, the fields touched by digitization and the fields rarely explored that suggest a future research agenda are discussed. The conclusions are formulated in the context of the specific objectives and the final document is elaborated.

## 3. Results and Discussion

### 3.1. Main Information about the Collection

#### 3.1.1. Summary of the Main Information

According to Table 1, the time space chosen for the analysis of this collection of 658 documents was 2018–2022. These publications contained 23,673 references; that is, an average of 35.98 references for each document. With regard to document type, the article led as the most prevalent type with 422 (64.13%), followed by the conference paper with 158 (24.01%) and the review with 29 (4.41%), allowing us to infer that this study was developed mostly from original documents that present recent research; that is, from primary sources. Finally, the collection included 1944 author keywords entered by the 2289 authors involved when indexing their documents.

**Table 1.** Summary of main information.

| Main Information about Data | |
| --- | --- |
| timespan | 2018–2022 |
| documents | 658 |
| references | 23,673 |
| article | 422 |
| book | 1 |
| book chapter | 9 |
| conference paper | 158 |
| editorial | 18 |
| erratum | 4 |
| note | 14 |
| review | 29 |
| short survey | 2 |
| keywords plus (ID) | 3390 |
| author keywords (DE) | 1944 |
| authors | 2289 |

### 3.1.2. Publications by Year, Country and Subject Area

Next, Figure 2 shows a graph divided into three parts, which highlights the publications for each year analyzed in this review, along with the 10 countries and 10 thematic areas with the most documents.

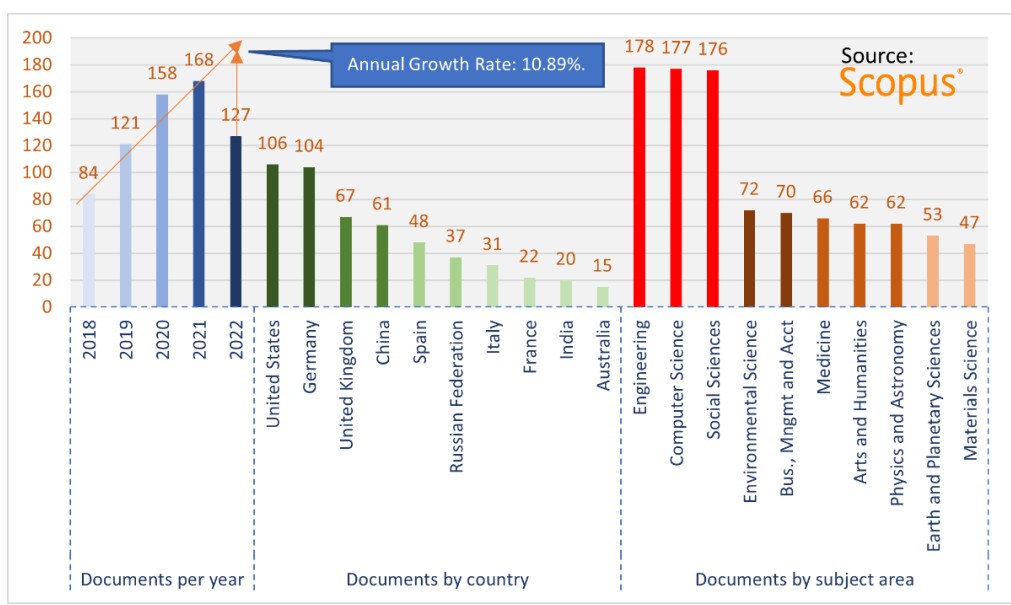

**Figure 2.** Publications by year, country and subject area.

With reference to the documents by year, 2018 indexed 84 investigations, followed by 2019 with 121 manuscripts and 2020 with 158 studies. Then, the year 2021 registered 168 documents, and at the date of the development of this review (September 2022) 127 manuscripts had been indexed in 2022; according to the annual growth rate of 10.89%, a projection of 186 documents by the end of 2022 is inferred.

In the case of the 10 countries with the most documents, the United States and Germany occupy first and second place with 106 and 104 documents, respectively. The United

Kingdom, China and Spain come in the third, fourth and fifth positions with 67, 61 and 48 separate investigations, respectively. Russia, Italy, France, India and Australia are positioned in sixth, seventh, eighth, ninth and tenth place with 37, 31, 22, 20 and 15 documents, respectively.

According to Figure 2, the areas of Engineering (n = 178), Computer Science (n = 177) and Social Sciences (n = 176) dominate as the top three fields with the most documents. With 60% fewer documents, Environmental Science (n = 72) and Business, Management and Accounting (n = 70) are in fourth and fifth place. Medicine (n = 66), Arts and Humanities (n = 62) and Physics and Astronomy (n = 62) rank sixth, seventh and eighth with 64% fewer documents relative to the top three. Finally, Earth and Planetary Sciences (n = 53) and Materials Science (n = 47) are in ninth and tenth place.

### 3.2. Fields Where Digitization Is Applied

The Thematic Map that is visualized in Figure 3 and that was generated by RStudio software is distributed in four blocks and each of these represents a category: Niche Themes, Motor Themes, Emerging or Declining Themes and Basic Themes. The topics reflected in this map are identified as clusters that orbit depending on the Degree of Relevance (Centrality) and the Degree of Development (Density). The default criteria used for the design of the Thematic Map are the author keywords (n = 250) and the minimum cluster frequency (n = 5) per thousand documents.

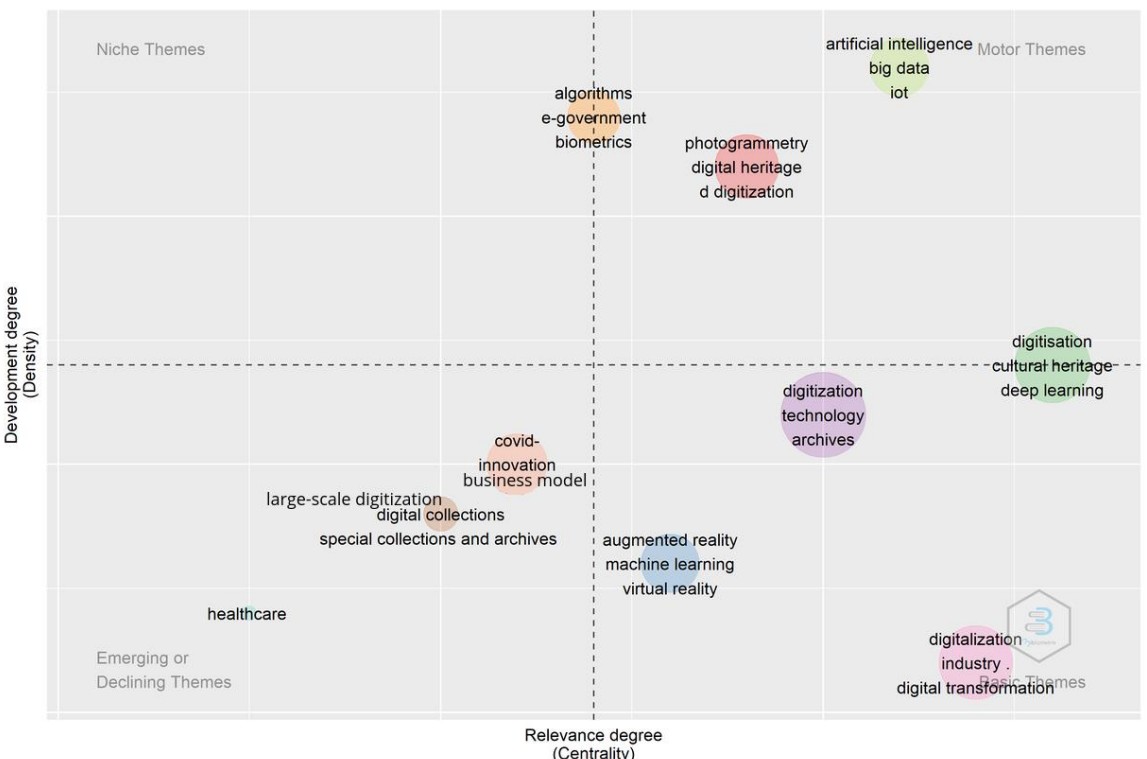

**Figure 3.** Thematic Map.

In accordance with the Thematic Map, it was possible to delimit the 10 fields where digitization is most often applied, and these are shown in Table 2.

**Table 2.** Fields identified in the Thematic Map.

| Field | Field Description | Field Category | Definition of the Category |
|---|---|---|---|
| 1 | "Digitization in Archives, Corruption and Economy" | Basic Themes | Issues caused by critical situations that affect our society |
| 2 | "Digitalization in Industry 4.0, Internet of Things, Sustainability and Big Data" | Basic Themes | |
| 3 | "Digitalization in Cultural Heritage, Deep Learning, Preservation and BIM" | Basic Themes | |
| 4 | "Digitization in Photogrammetry and 3D Digitization" | Motor Themes | Main themes of the research front |
| 5 | "Digitalization in Artificial Intelligence (AI) and Supply Chain Management" | Motor Themes | |
| 6 | "Digitalization in Augmented Reality, Machine Learning and Virtual Reality" | Basic Themes | |
| 7 | "Digitalization in Innovation, Business Model and Publishing Industry" | Declining Themes | Topics that are being left unstudied |
| 8 | "Digitalization in Algorithms, E-government and Biometrics" | Motor Themes | |
| 9 | "Digitization in Digital Collections" | Declining Themes | |
| 10 | "Digitalization and Healthcare" | Emerging Themes | New topics that are being studied more and more |

### 3.2.1. Field 1: Digitization in Archives, Corruption and Economy

According to Table 2, "Digitization in Archives, Corruption and Economy" is considered a Basic Theme and covers research on digitization with close links to archives, corruption and economy.

Regarding the digitization of files as a process of conversion from analog to digital format, there is an investigation called "The Archives at the Tip of Their Fingers: Exploring User Reactions to Large-Scale Digitization" prepared by [35], who conducted several interviews in order to analyze the responses of users of the digitization method "More Product, Less Process" at the University of Nevada, Las Vegas. The study found that digitization at a large-scale level positively impacted users; however, other strategies may be needed to make the most of the associated digital objects.

With regard to research related to digitalization as an anti-corruption strategy in private and especially public organizations, there is a document called "Outcomes of government digitization and effects on accountability in Benin" developed by [36], which aimed to specify incidences of digitalization reform in Benin to demonstrate that they can reduce corruption, connect citizens to administration and improve performance. Interviews were conducted with key external stakeholders and public employees. The findings showed that the application of new technologies has created setbacks for public employees and that reaching the total population remains a challenge. Nonetheless, the reform helped anticipate abuses and administration corruption, as well as create hopes for improved relations between the State and citizens.

Furthermore, this field includes publications such as the digitalization of the economy where digital technology is imposed on the economy, influencing the consumption, production, management and structure of organizations and leading to increasing automation and speed in the execution of tasks [37].

### 3.2.2. Field 2: Digitalization in Industry 4.0, Internet of Things, Sustainability and Big Data

According to Table 2, "Digitalization in Industry 4.0, Internet of Things, Sustainability and Big Data" is a Basic Theme and includes research on digitalization with close links to Industry 4.0, the Internet of Things (IoT), sustainability and Big Data.

The research related to Industry 4.0 deals with the digitalization of production processes in the industrial sector using information systems and sensors to make production processes more efficient. This is studied in the document "Impact of Industry 4.0 and Digitization on Labor Market for 2030-Verification of Keynes' Prediction" prepared by [38], where transitions in technological unemployment were examined and Keynes' theory was evaluated based on a study of the literature referring to the 4th Industrial Revolution. An analysis with a bibliographic approach was applied to 86 documents prepared between 2011 and 2020 related to Industry 4.0, technological unemployment, and the labor market. The research sample indicated that the impact of Industry 4.0 processes would reduce the volume of personnel needed, approaching Keynes' prediction of three (3) working hours a day.

Regarding research on the Internet of Things (IoT), which makes it possible for industries to automate and closely monitor business management to achieve fluidity and viability in costs and time, there is the article "Automation and digitization of agriculture using artificial intelligence and internet of things" by [30], which aimed to provide an overview of recent research in the agricultural sector induced by digital technology and the specification of a large number of applications using Artificial Intelligence (AI) and the Internet of Things (IoT). A review of several scientific databases was made, such as PubMed, WOS and Scopus, observing that the digitalization of the agricultural sector has evolved from its emerging conceptual phase to the application phase.

Sustainability research addresses how innovative solutions can have a transformative effect on individuals and collectives, conserving the environment and fostering a better quality of life. Among such research is a document entitled "Logistics and Agri-Food: Digitalization to Increase Competitive Advantage and Sustainability. Literature Review and the Case of Italy" developed by [39]. These authors study the obstacles that logistics must overcome in the agri-food industry. A review of the literature on the link between strategic management and logistics was carried out to obtain the level of growth in competitiveness in the agri-food industry. From this study, it emerges that logistics is an essential resource that provides competitive advantages and value to countries and companies; the ways to improve this resource are sustainability and connectivity.

Other research refers to digitalization and Big Data, which, when managed in a shared way, achieve results such as the prediction of behaviors and prediction of searches within a system [40,41].

### 3.2.3. Field 3: Digitalization in Cultural Heritage, deep Learning, Preservation and BIM

In accordance with Table 2, "Digitalization in Cultural Heritage, Deep Learning, Preservation and BIM" is considered a Basic Theme and covers digitalization research closely related to cultural heritage, deep learning, digital preservation and Building Information Modeling (BIM).

Research on cultural heritage addresses how digitalization assists in the preservation and conservation of heritage and scientific resources, creating educational opportunities and providing a solution to the world population's access to their heritage. One document on this topic is called "Methods of Digitization of Cultural Heritage. The Case Study of Terme di Diocleziano" developed by [42]; this study aimed to identify a unified survey methodology through which the elaboration of a digital advance work for archaeologies is viable. The monumental complex of the Terme di Diocleziano in Rome was taken as a case study.

Publications on deep learning deal with harnessing neural networks to improve tasks such as computer vision, speech recognition and natural language processing. One manuscript on this topic is entitled "A Deep Learning Digitization Framework to Mark up Corrosion Circuits in Piping and Instrumentation Diagrams" prepared by [43], who present a semi-automatic framework that makes it easier for operators to load a piping and instrumentation diagram without going through digitization so that connection points and pipeline specifications are located through deep learning. Then, by means of a heuristic

procedure, the text was obtained, oriented and read with the minimum degree of error, allowing the engineer to indicate the corrosion sections by means of a user interface.

Research on digital preservation studies the processes that are aimed at protecting the permanence of digital heritage documentation over time. An investigation around this topic is one called "Some Assembly Required: Low-Cost Digitization of Materials from Magnetic Tape Formats for Preservation and Access" developed by [44]; this research recommends that archivists and librarians without experience or technical skills consider the possibility of digitizing magnetic tapes individually. They describe in detail the challenges experienced by a team of inexperienced professionals in digitizing important audio recordings on open reel tapes and cassettes at the Northern Illinois University Library. The authors make it clear that the digitization of unstable and/or deteriorated materials in magnetic tapes contribute to the conservation of their information.

Building Information Modeling (BIM) research examines how the use of BIM creates and manages data throughout the design, construction and operations process. BIM can create digital representations by integrating multidisciplinary data that are then managed on a cloud platform to ensure in-person collaboration. On this topic there are two investigations elaborated by [45,46].

### 3.2.4. Field 4: Digitization in Photogrammetry and 3D Digitization

According to Table 2, this field is indicated as being a Motor Theme and research on digitization is identified with a close relationship to photogrammetry and 3D digitization.

Research related to photogrammetry deals with studies that use techniques that facilitate the obtainment of reliable data of practical things in the environment through the development of 3D models based on photographs. An article linked to this topic is called "Automating Photogrammetry for the 3D Digitisation of Small Artifact Collections" presented by [47], who propose a modern scanner for small objects that manages to automate the procedure of taking images through photogrammetry to achieve a very complete and effective 3D digitization of cultural heritage materials in important museum collections.

Publications on 3D scanning deal with how replicas of objects can be created in a digital format and then archived, modified or reproduced permanently. A study on this topic is entitled "Low-Cost Prototype to Automate the 3D Digitization of Pieces: An Application Example and Comparison" carried out by [48], which aimed to present the model of a technical and easy programming 3D capture system for later use with a DSLR camera or 3D scanner using photogrammetry. The authors hope that this design will overcome the limitations of existing equipment on the market (rotary tables or robotic arms). With regard to its manufacture, FDM additive technology, structural components available on the market, the firmware of 3D printers (Arduino), a Spider Artec 3D scanner and a Nikon 5100 SLR camera were used. Positive results were obtained when developing 3D models contrasting the 3D meshes achieved by the two methods.

### 3.2.5. Field 5: Digitalization in Artificial Intelligence (AI) and Supply Chain Management

According to Table 2, this field is considered a Motor Theme and covers research on digitalization with close links to Artificial Intelligence and supply chain management.

The documents on Artificial Intelligence address content on the automation of productivity operations in organizations to increase their profitability. AI is achieved with a combination of significant volumes of data, programming and machine learning. Consequently, it allows machines to respond in a precise way in the face of an incident and to be designed so that they learn to respond correctly on their own. One document on this topic is called "Social Perception of Artificial Intelligence and Digitization of Cultural Heritage: Russian Context" prepared by [49], which aimed to understand from a theoretical point of view the digital ontology and implementation of AI in the sense of the Russian "Realia". This study executed a complete analysis with the support of statistical data, using comparative and descriptive methodology. It also explained the objective and subjective reasons for negative opinions of digital devices, considering the roles of important characters in

digital ontology (data scientists, influencers and stakeholders). This research reveals three factors involved in the frontiers of digitalization: the axiological factor, the vector subject and the ethical factor.

The documents on supply chain management deal with how digitalization ends the obstacles between processes, allowing the supply chain to become a complete and open ecosystem for all involved (suppliers and the manufacturers of finished products or end users). This topic is highlighted in research called "Supply Chain Digitisation Trends: An Integration of Knowledge Management" developed by [50], whose purpose was to understand the possible consultations of teachers in order to enhance their horizons and take advantage of knowledge management in order to deepen the research model on the digitalization of the supply chain. A literature review was conducted, along with a textual analysis and estimates on applications, digitization issues and technologies in the field and industry. Data from Google trends in the period from 2010–2018 were contrasted in two measures (growth and prevalence) to establish differences between media (videos and news) and academic publications, weighing academic performance with respect to the results in the previously indicated areas of the digitalization of the supply chain.

### 3.2.6. Field 6: Digitalization in Augmented Reality, Machine Learning and Virtual Reality

"Digitalization in Augmented Reality, Machine Learning and Virtual Reality" is considered a Basic Theme according to Table 2 and addresses research on digitalization with strong links to Augmented Reality (AR), Machine Learning (ML) and Virtual Reality (VR).

AR as a technology makes it possible to superimpose virtual components above our perception of things. One publication that stands out on this topic is called "A Digital System for AR Fabrication of Bamboo Structures through the Discrete Digitization of Bamboo" prepared by [51], where the design of a methodology for assembling bamboo canes from a set of instructive mobile algorithms is presented, supported in applied AR and ML procedures and a material analysis. The method was evaluated in numerous tests and the system was well functioning; consequently, the addition of solid structures according to the allocated resources was proposed. These results presuppose great benefits if assemblies of family units were carried out by unqualified personnel using automated equipment.

ML is a branch of Artificial Intelligence (AI) that is focused on designing machines that manage to learn or raise productivity, subject to the data used. On this topic, there is a manuscript entitled "Digitization of Broccoli Freshness Integrating External Color and Mass Loss" by [52], which proposes a novel index to evaluate the freshness in green vegetables combining the loss of mass and the degree of greenness. The mass retention rate was calculated by weighing while the green color retention rate was calculated using a computer vision system. The findings showed that the novel evaluation index composed of mass and greenness is more detailed than the traditional index which uses only greenness.

VR can be considered a context of situations and things with realistic aspects that manages to create in the user the impression of being immersed within. The environment is usually observed by means of glasses or a VR headset. Regarding this topic, a document called "Virtual Reality-Based Digitisation for Endangered Heritage Sites: Theoretical Framework and Application" developed by [53] was drafted, which aimed to design an integrated framework digitally using VR technology; this was used in a digital endorsement for the creation of a simulated scenario of compromised heritage sites and consequently to address how recent urbanisms could harm their presence. The study applied a qualitative longitudinal method, and the designed framework was validated by field data collection from a case study on the ancient cone-shaped community settlement in Kardan, Iran. According to the authors, the results could raise awareness, promote will and challenge the status quo about this heritage establishment during an engaging and interactive presentation to the public.

### 3.2.7. Field 7: Digitalization in Innovation, Business Model and Publishing Industry

According to Table 2, this field is considered a Theme in Decline and includes publications on digitalization with a strong relationship to innovation, business models and the publishing industry.

Digital innovation is about the development of a business model or value proposition through digital equipment. That is why business models must have a strong link with new technologies. On this topic, the manuscript entitled "Spatial Implications of Digitization: State of the Field and Research Agenda" prepared by [54] stands out, which aimed to focus on the economic geographical implications of digitalization and expand the available literature in two ways. First, this study examined the reality of research about the geographical results of digitalization. Second, a future research agenda was developed on topics that could be studied by economic geographers, such as the effects of spatial economic digitalization and digital skills, among others.

With regard to the digital business model, such a model takes advantage of technology to strengthen services within a company, with its suppliers, partners, and end users awarded additional benefits, but it also makes the service or product offered to the market profitable. This topic is covered by the article entitled "Digitization Capability and the Digitalization of Business Models in Business-to-Business Firms: Past, Present, and Future" developed by [55], which aimed to create a global approach and strengthen the body of research based on a historical summary report regarding "research on digitalization" and "digitalization in business-to-business markets", inferring that this debate is traditional; therefore, it is not a new problem. These authors developed a concept of digitization capacity as a criterion to discuss to what extent the digitization capacity of an organization collaborates with its business model, facilitating expansion with the help of information, specifically, its digitization.

With the advent of the digitization of information, ways of reading hardly changed. Neither did the process of making books. The publishing industry is an industry that has adapted well to change. Digitalization resulted in novel content distribution networks, such as social networks, which had great power with regard to disseminating content from publishers to a segmented audience. On this topic, the document called "The Impact of Digitization in Spanish Scholarly Publishers" by [56] stands out, which analyzes the incidence of digitalization in the Spanish publishing industry. The current document had two fundamental objectives: The first was to establish the degree of magnitude of the repercussions of digitalization on the Spanish publishing industry, and the second was to identify the presence of recent business models in the publishing industry in Spain. An empirical analysis was carried out through surveys aimed at a population extracted from a sample carried out on Spanish academic publishers. Stratified random sampling was applied as a criterion.

### 3.2.8. Field 8: Digitalization in Algorithms, E-Government and Biometrics

Based on Table 2, this field is considered a Motor Theme and involves publications where digitization has a close relationship with algorithms, e-government and biometrics.

An algorithm is a process that is given gradually to ensure an outcome. Based on an original criterion and data, a set of ordered steps is applied to solve a problem. In computer programs, an algorithm consists of the first stage for writing code. This topic includes research entitled "Automatizing Chromatic Quality Assessment for Cultural Heritage Image Digitization" carried out by [57], which dealt with an obstacle in the evaluation of image quality (IQA) in reference to the digitization of cultural heritage, resorting to machine learning (ML). The authors analyzed the option of designing a decision tree that replicates the result of a specialist when visualizing images of cultural assets.

E-government, also known as digital government, uses communication and information technologies to enable governments to approach voters, enhance services, be more productive and gradually be more involved with new sectors of society. On this subject, [58] prepared a document entitled "Venezuelan State: Lack of Digitization as Concealment"

where he pointed out how in the last two decades, the negligence of state agencies has been allowed and there has been little clarity with regard to rules, in addition to centralism to the detriment of the digital government and accountability. In the case of Venezuela, the lack of digitalization, except for a few processing agencies and benefactor offices after 2015, is an implicit factor at the service of the invisibility of information to the public, being a problem that adds to the rampant corruption experienced by a large number of Venezuelans.

Biometrics is an automated program and is oriented towards the recognition of the physical aspects of people that are non-transferable and exclusive, including the fingerprint, face or iris, and is considered to verify a person to avoid fraud. The authors of [59] developed an investigation on this topic called "An Inquiry into the Digitisation of Border and Migration Management: Performativity, Contestation and Heterogeneous Engineering", which describes three implications of digitalization. They claim that the Visa Information System (VIS) compiles a group of hitherto isolated government authorities into a set of end users who approve migration management and border security through data collection, generation and trafficking; enables tracking for the control of displacement; and has negative impacts on migrants' abilities to manage and deal with control.

The paper also incorporates three analytical sensitivities that guarantee the prevention of analytical traps when exploring digitization processes. First, it refers to how security procedures analyze the identities of the users they are addressing; then, it considers possible practices of subversion on the part of migrants to avoid unbalanced control analyses; finally, it describes the challenge of considering the development of border security technologies.

### 3.2.9. Field 9: Digitization in Digital Collections

According to Table 2, this field is considered a Theme in Decline and encompasses research on digitization with a strong link to digital collections. What is a digital collection? It can be said that a digital collection is based on digital resources that have gone through a selection and organization process to facilitate their accessibility and operation, depending on the policy of the materialization of the collection, which previously needs to be agreed upon and documented to begin its digitization. Digital collections are managed through digital repositories, which are responsible for storing, sorting, categorizing and providing the user with the digital elements that constitute them.

The authors of [60] produced a manuscript on this topic, which is entitled "Cultivating Digitization Competencies: A Case Study in Leveraging Grants as Learning Opportunities in Libraries and Archives." This case study explains how six (6) digitization fields were developed and publicized as part of donation-funded projects at the University of Nevada Special Collections and Library Archives, Las Vegas. These fields are "donation writing", "project planning", "metadata", "project management", "digital asset management" and "digital capture". The authors presented the fields separately, discussed their importance and explained both how it was developed over the course of the grant project and how it was possible to teach in a workshop setting. Finally, discrepancies regarding competency achievement within three independent groups of participants were discussed: (1) newly graduated human resource workers with a grant who eventually gain experience; (2) library specialists in digital collections who experiment and innovate; and (3) public officials specializing in cultural heritage who are part of workshops sponsored by donations.

### 3.2.10. Field 10: Digitalization and Healthcare

According to Table 2, this field is an Emerging Theme, and includes publications on digitalization with a strong relationship to the health sector. Digital health or e-health is composed of a package of digital equipment used in the prevention, diagnosis and treatment of diseases, together with the supervision of the affected and the management of health.

One investigation on this topic is called "Digitalization of the Health Sector in Pakistan: Challenges and Opportunities to Online Health Communication: A Case Study of MARHAM Social and Mobile Media" developed by [61]; this study aimed to examine the

role of social networks and mobile media in the digitization of the healthcare industry in Pakistan using the MARHAM platform. The findings of the study affirm that the platform plays a significant role in the health area of Pakistan, solving problems for children and women thanks to its Facebook group. They concluded that MARHAM faces many challenges akin to most online platforms, including limited internet access and a low rate of public training.

### 3.3. Future Research Agenda

After a discussion of the 658 documents selected for this review, together with its 3390 plus keywords and 1944 author keywords (Figure 4), it was possible to extract eight fields rarely explored in digitization. These eight fields are: "Automated and Connected Driving", "Digital Aggregators", "Digitization of Higher Education", "Digitization of Agriculture", "Digital Dictionaries", "Color Management System", "Digitization in Customs Payment" and "Digitization of Chess Moves".

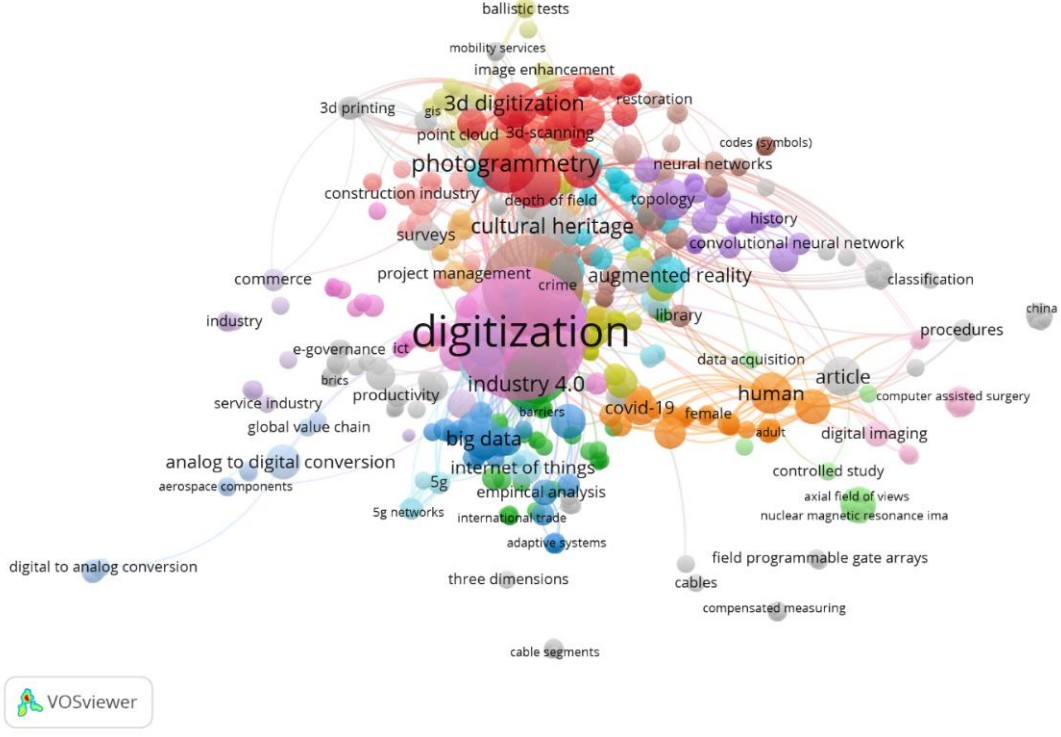

**Figure 4.** Author keywords.

The selection criterion of these eight fields was to search in each cluster for the keyword with the least co-occurrence, that is, a variable rarely studied but linked to digitalization. These keywords were: "Automated and Connected Driving" (n = 2), "Aggregators" (n = 2), "Higher Education" (n = 2), "Agriculture" (n = 2), "Digital Dictionaries" (n = 2), "Color Management System" (n = 2), "Customs Payment" (n = 2) and "Chess Moves" (n = 2).

Next, we discuss the eight little explored fields:

- Automated and Connected Driving: Studies in this field are aimed at reducing fatal road accidents, harmful emissions from transport and traffic [62].
- Digital Aggregators: Field of study on tools designed for musical groups and artists to upload their releases to platforms and/or digital stores [63].
- Digitization of Higher Education: Field of study that deals with the education of individuals who are disseminated from a geographical point of view or connect with teachers via the Internet in deferred time [64,65].

- Digitalization of Agriculture: Field of study that deals with the tangible possibility of carrying out healthier, sustainable and more inclusive food structures, but not without risks [66].
- Digital Dictionaries: Field of study that deals with computer applications that consider a database enriched with content and formats (image, text, video or sound) accompanied by hardware for consultation, easy use and attractive visualization [67].
- Color Management System: Field of study that deals with the formulations supplied for standardized color duplication for teams involved within the digital workflow [68].
- Digitization in Customs Payment: Field of study that deals with the implementation of effective processes in customs activities, that is, fast transactions and accountability during the entry–exit of cargo [69].
- Digitization of Chess Moves: Field of study oriented towards the assisted digital operation of a chess game by means of images of score sheets and with the purpose of officially maintaining records of chess events [70].

## 4. Limitations

First, this bibliometric review is limited to a single database called Scopus. Second, the search was limited to open access documents. Finally, its results are only up to date as of September 2022.

## 5. Conclusions

To conclude, this study showed that digitalization has conquered many areas and is here to stay. According to the key information obtained, 88.14% of the documents selected for this bibliometric review comprise original research, allowing us to infer that the results are based on valuable information. In terms of the production of documents per year, the results report an annual growth rate of 10.89%; that is, research is constantly growing because more and more fields of study are taking advantage of digitalization.

The United States leads document production, with 106 manuscripts published from 2018–2022, followed by Germany with 104. The production of manuscripts in Germany is striking compared to that of the United States, and this is because lately the governors of Germany have agreed that the German people need a deep modernization to take place in their economy, but without neglecting the State. They plan to head towards a decade of digital relaunch.

Engineering, Computer Science and Social Sciences are the leading fields of knowledge, with 178, 177 and 176 documents published. This similarity in document publication rates allows us to infer how linked these three areas are, and that digitization technologies need the participation of the Social Sciences for the progress and consolidation of society.

To respond to RQ1, it was possible to identify 10 fields of study where digitalization is most applied, of which four are Basic Themes, three are Motor Themes, two are Declining Themes and one is an Emerging Theme (Table 2). After debating what fields of study could possibly make up a future research agenda, it was possible to derive eight areas so far little explored.

It is suggested that research is carried out in the fields categorized in Table 2 as a Theme in Decline and Emerging Theme, and in the eight fields indicated in the future research agenda. This review is to date one of the most up-to-date documents on digitization (September 2022).

These findings show the ten most explored fields from 2018–2022 and propose another eight (little explored) for future studies. Therefore, this study is useful for future research related to digitalization because it indicates numerous fields to explore.

**Author Contributions:** Conceptualization, L.E.-R. and J.G-A.; methodology, L.E.-R.; software, L.E.-R.; validation, L.E.-R. and J.G.-A.; formal analysis, L.E.-R.; investigation, L.E.-R.; resources, J.G.-A.; data curation, J.G.-A.; writing—original draft preparation, L.E.-R.; writing—review and editing, L.E.-R.; visualization, J.G.-A.; supervision, J.G.-A.; project administration, L.E.-R. All authors have read and agreed to the published version of the manuscript.

**Funding:** This research received no external funding.

**Institutional Review Board Statement:** Not applicable.

**Informed Consent Statement:** Not applicable.

**Conflicts of Interest:** The authors declare no conflict of interest.

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
