# Peer review of "Fields Touched by Digitalization: Analysis of Scientific Activity in Scopus"

_sustainability, doi:10.3390/su142114425_

Round 1
Reviewer 1 Report
The authors presented a relatively interesting study for consideration, in which, however, the relatively brief introduction ends with a relatively general question. At this point, I would like to add one more question, namely: How is the thematic axis of the presented study related to the main thematic axis of the journal? Perhaps I will add a sub-question: Does the publisher have thematically compatible journals in its portfolio? The answer to the first is: very little if at all, the answer to the second is: Yes, quite a significant amount.
At this point, I will end my assessment and suggest a reconsideration of the choice of a journal that thematically corresponds to the chosen field of science. In case the authors insist on a specific journal, I recommend choosing a corresponding special issue that will thematically match the content. Subsequently, I suggest resubmitting the study for assessment.
Author Response
Consulte el archivo adjunto.

Reviewer 2 Report
Thank you for the opportunity to review this paper. This is a well written and interesting article. I read the manuscript carefully. The manuscript aims to answer what the fields of study where digitalization is most applied are. The paper addresses a topical issue and makes a potentially interesting contribution. I found it to have potential as an article for Sustainability, and I suggest below how I think it might be improved.
1. Whether the data collection has gone through a strict process, such as the search of this keyword and the filtering of the data, and whether the irrelevant article has been filtered out.
2. Whether there is a discussion on this related field instead of the scientific map of light and light, and whether there is a discussion on the future research direction.
3. Introduction is not well written, and there is a related study focused on digitalization and bibliometrics: https://doi.org/10.1016/j.techfore.2021.121037; https://doi.org/10.1002/sd.2242. you'd better discuss this paper for comparison.
Author Response
Por favor vea el archivo adjunto

Reviewer 3 Report
The subject is very interesting and in general is very well discussed.
But in my opinion, I think that it is necessary to take into consideration the next recommendations:
P1: The Abstract must contain very clear the goal of your research. In general, in an Abstract is avoided to start with a question.
P2: We identified an Introduction without an objective and without a clear literature review on the subject proposed to research. I recommend to include a clear literature review regarding the research question assumed by the authors.
P3: In the theory, there is a lot of questions of research recommended for this bibliometric analysis. Please, analyze the preview works in the matter and motivate your research question from this perspective. Please, avoid the redundancy approach!
P4: There is no motivation for the next aspects: period of time (the first digital economy concept was launched in 1996); the field of research (why LIMIT-TO (SUBJAREA, "COMP") OR 93 LIMIT-TO (SUBJAREA, "BUSI") OR LIMIT-TO (SUBJAREA, "ARTS") ?), and what is the link with research question?
P5: I don't understand very clearly the model of research: you started from a research question RQ1, after you mentioned the Zupic & Carter model (but you didn't use this model), after a simple distribution analysis you used R-Studios for identified the directions of digitizations, you analyzed in a similar way as SLR (systematic literature review) every direction identified,
P6: The approach mentioned in the methodology is not respected, finally. There is section 2.3 is without an own subject.
P7: Section 3.3 is not clearly explained. We cannot deduce from the text if we have a co-occurrence keyword or another type of co-occurrence. Table 3 and Figure 4 are not discussed in the context of this bibliometric analysis.
P8: The section "Conclusions and Recommendation" is not appropriate. It is necessary to respect the standard structure for Conclusions. Please, separate the Conclusions !
P9: There are no limitations discussed for this approach.
P10: We didn't identified the motivation and utility of this study.
P11: This study is not a full Bibliometric analysis nor a Systematic literature review.
Author Response
Por favor vea el archivo adjunto

Round 2
Reviewer 1 Report
My reservations from the original review remain, I leave it to the editors to decide whether to publish/reject.
Author Response
"Por favor vea el archivo adjunto"

Reviewer 3 Report
In my actual review, I will analyse the level of implementation for the issues from my preview review.
P1: The Abstract must contain very clear the goal of your research. In general, in an Abstract is avoided to start with a question.
Ok.
P2: We identified an Introduction without an objective and without a clear literature review on the subject proposed to research. I recommend to include a clear literature review regarding the research question assumed by the authors.
Ok. The reference style is embeded by using only "[]" and a little dificult to understand.
P3: In the theory, there is a lot of questions of research recommended for this bibliometric analysis. Please, analyze the preview works in the matter and motivate your research question from this perspective. Please, avoid the redundancy approach!
Ok.
P4: There is no motivation for the next aspects: period of time (the first digital economy concept was launched in 1996); the field of research (why LIMIT-TO (SUBJAREA, "COMP") OR 93 LIMIT-TO (SUBJAREA, "BUSI") OR LIMIT-TO (SUBJAREA, "ARTS") ?), and what is the link with research question?
Partial. The motivation is not clear for period 2018-2022. The string for interogation is diferrent in current version in comparation with the previous version, but is not explained "why?".
P5: I don't understand very clearly the model of research: you started from a research question RQ1, after you mentioned the Zupic & Carter model (but you didn't use this model), after a simple distribution analysis you used R-Studios for identified the directions of digitizations, you analyzed in a similar way as SLR (systematic literature review) every direction identified,
Partial: The part with model is very simple presented. Is not clear where is included the visualization as a step of any bibliometric analysis.
P6: The approach mentioned in the methodology is not respected, finally. There is section 2.3 is without an own subject.
Ok, but the methodology is very simple.
P7: Section 3.3 is not clearly explained. We cannot deduce from the text if we have a co-occurrence keyword or another type of co-occurrence. Table 3 and Figure 4 are not discussed in the context of this bibliometric analysis.
In the text there is not Table 3, at this moment.
P8: The section "Conclusions and Recommendation" is not appropriate. It is necessary to respect the standard structure for Conclusions. Please, separate the Conclusions !
Ok.
P9: There are no limitations discussed for this approach.
Ok.
P10: We didn't identified the motivation and utility of this study.
Ok. We prefer to see motivation in the first part of paper, and not in Conclusions.
P11: This study is not a full Bibliometric analysis nor a Systematic literature review.
Partial.
General conclusion: We identified only a simple conformation at the our recommendations. See the model of research! The qualitative sault is not so present in this version.
Author Response
"Por favor vea el archivo adjunto"